# Higher Antibody Concentrations in U.S. Health Care Workers Associated with Greater Reactogenicity Post-Vaccination

**DOI:** 10.3390/vaccines10040601

**Published:** 2022-04-13

**Authors:** Jane A. Dickerson, Janet A. Englund, Xing Wang, Julie C. Brown, Danielle M. Zerr, Bonnie Strelitz, Eileen J. Klein

**Affiliations:** 1Seattle Children’s Research Institute, Seattle, WA 98145, USA; janet.englund@seattlechildrens.org (J.A.E.); xing.wang@seattlechildrens.org (X.W.); julie.brown@seattlechildrens.org (J.C.B.); danielle.zerr@seattlechildrens.org (D.M.Z.); bonnie.strelitz@seattlechildrens.org (B.S.); eileen.klein@seattlechildrens.org (E.J.K.); 2Laboratory Medicine and Pathology, University of Washington School of Medicine, Seattle, WA 98145, USA; 3Division of Infectious Diseases, Department of Pediatrics, University of Washington School of Medicine, Seattle, WA 98145, USA; 4Division of Emergency Medicine, Department of Pediatrics, University of Washington School of Medicine, Seattle, WA 98145, USA

**Keywords:** SARS-CoV-2, vaccine reactions, antibody measurement

## Abstract

Multiple factors may be associated with immune responses to SARS-CoV-2 vaccines. Factors potentially related to magnitude and durability of response include age, time, and vaccine reactogenicity. This study analyzed SARS-CoV-2 IgG spike antibody responses following the second dose of vaccine in healthcare workers (HCWs). Data were collected from participants enrolled in a longitudinal SARS-CoV-2 serology study over a 12-month period. Participants completed a survey documenting symptoms post-vaccination. Serum specimens were tested for SARS-CoV-2 IgG antibodies using the Abbott Architect AdvisdeDx SARS-CoV-2 IgGII assay. Antibody levels were compared against time from second vaccine dose, and symptoms following vaccination. Altogether, 335 women (86.6%) and 52 men (13.4%) participated. Median age was 37 years (IQR 30-43). Overall median antibody level was 2150.80 [1246.12, 3556.98] AU/mL (IQR). Age was not associated with antibody concentration (*p*-value = 0.10). Higher antibody responses (2253 AU/mL vs. 1506 AU/mL; *p* = 0.008) were found in HCWs with one or more symptoms after the second dose of the vaccine (n = 311). Antibody responses persisted throughout the study period post-vaccination; statistically significant decreases in antibody responses were observed over time (*p* < 0.001). Higher antibody response was associated with reactogenicity post-vaccine. Age and sex were not associated with higher antibody responses.

## 1. Introduction

In response to the COVID-19 pandemic, unprecedented efforts have been made to rapidly develop and distribute effective vaccines on a global scale. Since the first COVID-19 vaccines received emergency use authorization (EUA) for adults in the United States in December 2020, more than 5 billion people and over two thirds of the world population have received at least one dose of a COVID-19 vaccine [1]. As of March 2022, three vaccines have either full approval or EUA from the Food and Drug Administration (FDA) in the United States, including Pfizer’s BNT162b, Moderna’s mRNA-1273, and Johnson & Johnson’s Janssen vaccines. Clinical phase III trials have demonstrated excellent efficacy, although the trials focused more on assessing the outcome of preventing symptomatic disease, and less on characterization of the immune responses [2,3,4]. There is increasing interest in understanding the durability of immune responses post-vaccination, as well as predicting levels of immune responses in different populations as booster doses become approved for adults and endorsed by regulatory bodies in the USA and elsewhere.

Several commercial platforms have received EUA from the U.S. FDA for the laboratory assessment of humoral response to COVID-19. While variation among commercial platforms exists, literature suggests there is good qualitative correlation among different vendors and target antigens [5]. Newer evidence suggests antibody response over time is assay dependent [6,7]. A few studies have demonstrated good qualitative performance using existing automated assays, such as Abbott’s AdviseDx SARA-CoV IgII, Roche’s Elecsys^®^ Anti-SARS-CoV-2 S, Diasorin’s LIAISON^®^ SARS-CoV-2 S1/S2 IgG, as well as GenScript’s FDA approved neutralization test [8,9,10].

Antibody response following immunization with COVID-19 vaccines has been investigated using a variety of commercial platforms with EUA or lab-developed tests. These studies demonstrate robust, early immune responses. An Israeli study demonstrated SARS-CoV-2 receptor-binding domain IgG antibodies were present in 99.9% of 1487 vaccinated health-care workers tested two weeks after a second Pfizer-BioNTech BNT162b2 vaccine dose, and neutralizing antibodies were present in 96.5% of 357 participants in an enriched subgroup with comorbidities tested one week after the second dose [11]. An Italian study evaluated immune response seven days post second dose of BNT162b and correlation with age, sex, and BMI, and found a significant difference in titers between male/female and age groups [12]. A few reports have documented the durability of immune response for up to three months, although a statistically significant decline in antibody levels over time has been noted even during this short time frame, which raises concern about long term immunity [8,13,14,15]. Another Israeli study reported continual decrease in spike antibody concentration over six months in a large cohort of 4868 healthcare workers [16]. More data are needed to assess the longer-term durability and correlates of protection post-vaccination in diverse populations, including the relationship of vaccine reactogenicity and protection. Vaccine reactogenicity includes side-effects commonly observed after vaccination (e.g., fever, headache).

This study reports on the humoral response to mRNA vaccines in SARS-CoV-2 naïve healthcare workers (HCWs) using the Abbott AdviseDx SARS-CoV-2 Spike IgG assay. The goals of the study were to analyze the SARS-CoV-2 spike protein antibody concentration after the second dose of vaccine in a group of 387 healthcare workers (HCWs), and to assess correlation of antibody concentration with age, vaccine reactogenicity, and time since vaccine.

## 2. Materials and Methods

This study was completed at an academic, tertiary care pediatric hospital in Seattle, WA, U.S. Data were collected from participants enrolled in a longitudinal SARS-CoV-2 IgG serology study over a 12-month period (May 2020–May 2021), as approved by the Seattle Children’s Hospital Institutional Review Board. A description of the enrollment process and inclusion criteria was previously reported [17]. Briefly, employed HCWs, including physicians, nurses, pharmacists, administrative staff, social workers, mental health evaluators, security team members, child life specialists, and housekeepers, were recruited if they worked at least one shift in a clinical setting 14 days prior to enrollment, and were not experiencing any COVID-19 related symptoms. Blood samples were collected longitudinally in May 2020, August 2020, November 2020 and May 2021. Participants were eligible for vaccination beginning in December 2020. At the final blood draw, specimens were tested for a vaccine associated spike protein IgG in addition to IgG against SARS-CoV-2 nucleoprotein tested at all four blood draws. This study focuses on testing done at the fourth blood draw.

Frontline health care and hospital workers completed a short survey at the time of each blood draw to collect demographic information, current and recent symptoms, hospital role, typical work location, known exposures, previous COVID-19 diagnosis, and medical history, including immunosuppressant medication use or chronic underlying conditions (Table 1). The mRNA COVID-19 vaccines became available to HCWs through a hospital-based vaccine program beginning December 2020. At the time of the 12-month post-enrollment blood draw, participants were also asked to complete a survey, based on CDC’s v-safe survey [18], documenting the number and severity of symptoms experienced after the second dose of the vaccine (Table 2).

Participants provided up to four serology specimens at four time points, with only one post-vaccination serology specimen obtained at the last time point in the longitudinal study one year after enrollment. Because participants were vaccinated at different times, the post-vaccination specimens could have been obtained anywhere from one to six months after vaccination. Approximately 6 mL of blood was collected from each participant, allowed to clot at room temperature for ~1 h, and refrigerated and centrifuged within 24 h.

Antibody levels were stratified into five groups based on the time after the second vaccine dose that the sample was obtained: 30 to <90 days (n = 12, median 85.5 [74.3, 87.0] days), 90 to <120 days (n = 44, median 113.0 [108.0, 115.3] days), 120 to <150 days (n = 159, median 130.0 [127.0, 134.5] days), 150 to <180 days (n = 89, median 173.0 [165.0, 176.00 days) and 180 to 210 days (n = 59, median 184.0 [182.0, 187.0] days).

Serum specimens were tested for SARS-CoV-2 IgG antibodies using the Abbott Architect AdviseDx SARS-CoV-2 IgGII assay and SARS-CoV-2 IgG Assay. AdviseDx is a quantitative test for IgG against the SARS-CoV-2 receptor binding protein (RBD). An antibody level >50 AU/mL was considered positive based on the manufacturer’s instructions. The expected imprecision for the test is <5% coefficient of variation [8]. The sensitivity of the Abbott Architect AdviseDx SARS-CoV-2 IgGII assay 10 days after positive polymerase chain reaction (PCR) testing results has been reported to be 95.6%, with specificity of 100% [7]. The SARS-CoV-2 IgG assay is a qualitative test for IgG against the SARS-CoV-2 nucleoprotein. An antibody index >1.4 was considered positive, using the manufacturer’s instructions.

Demographic characteristics were described by means, medians, IQR, and ranges for continuous variables, and frequencies and percentages for categorical variables. A Mann–Whitney U test or Kruskal–Wallis test was performed for evaluation of factors associated with positive SARS-CoV-2 IgG serology testing results. Age was modeled as a continuous variable. The percentages of SARS-CoV-2 IgG Spike antibody reduction were evaluated using a linear model with log-transformed SARS-CoV-2 IgG Spike results. All statistical analyses were performed using R Statistical Software (version 3.6.1; R Foundation for Statistical Computing, Vienna, Austria).

## 3. Results

Altogether, 335 women (86.6%) and 52 men (13.4%) participated in this study (Table 1). Participants had a median age of 37 years (IQR 30-43). Most participants (363, 93.8%) received the Pfizer BNT162 COVID-19 vaccine, and the remainder received Moderna’s mRNA-1273 vaccine. At the time of the blood draw, participants ranged from 50 to 193 days after their second vaccine. All participants had a positive spike antibody response, defined as >50 AU/mL. The overall median antibody level was 2150.8 [1246.1, 3557.0] AU/mL (IQR). Age was not associated with SARS-CoV-2 IgG Spike antibody concentration (spearman correlation coefficient = −0.083, *p*-value = 0.10). Sex at birth was not associated with SARS-CoV-2 IgG spike antibody concentration (*p*-value = 0.65).

### 3.1. Post-Vaccine Reactogenicity

Reactogenicity experienced immediately after the second dose of the vaccine was reported by 311 (80.4%) participants at the 4th blood draw survey, a median of 136 days post-vaccine dose 2 (range, 50–193 days). Antibody levels of greater magnitude were observed in HCWs who reported experiencing one or more symptoms after the second dose of the vaccine (n = 311). This difference (2253 AU/mL vs. 1506 AU/mL) was statistically significant (*p* = 0.008; Figure 1). The symptoms most strongly associated with increased antibody concentration in our study were fatigue/tiredness, fever/chills, and body aches (Figure 1).

### 3.2. Antibody Levels over Time

Antibody levels persisted in all participants, ranging from 1 to 7 months post-vaccination at the time of their blood draw. A statistically significant decrease in antibody concentration was observed over time using a linear model with log-transformed responses (*p* < 0.001, Figure 2).

## 4. Discussion

The durability of vaccine-associated antibody responses are important indicators of long-term vaccine-associated immunity [19,20]. In this study, we examined the durability of SARS-CoV-2 IgG Spike antibody response in 387 HCWs from 50 to 193 days after the second mRNA SARS-CoV-2 vaccination. Every participant demonstrated sustained IgG responses to the viral spike protein with respect to the assay’s pre-specified measure of positivity (>50 AU/mL). Other studies have reported good correlation between the Abbott AdviseDx assay and surrogate virus neutralization tests, suggesting that humoral antibody titers correspond well to the presence of neutralizing antibodies [8,9]. Although no participants were seronegative, we report a marked and statistically significant decrease (*p* < 0.001) in antibody concentrations detected over time, with levels ranging from medians of 9496 AU/mL down to 1808 AU/mL. The antibody levels decreased by 48% (90–119 days), 64% (120–149 days), 66% (150–179, and 78% (180–193 days) in the five measured time periods over seven months post-vaccination. This finding is consistent with other studies, although these data notably represent one of relatively few reports that describe antibody concentrations beyond three months post-vaccination [16,21,22]. In addition, we had very high continued participation throughout the study (73% of the original cohort tested at the fourth visit) compared with other published studies, resulting in decreased opportunity for the introduction of bias in the population tested. Levin et al. also reported a continual decrease in spike antibody concentration over six months in a larger cohort of 4868 healthcare workers, although only approximately one third of subjects (1370) were tested at the final visit compared with the baseline visit (3991) [16]. Similarly, to this study, there was a steady decline in antibody levels throughout the six- month study period, with few subjects falling below the antibody cutoff level for a positive test.

Two studies have demonstrated an association between SARS-CoV-2 antibody levels and immune protection. In a study by Khoury et al., the authors used published data from seven vaccine studies to determine the log mean of neutralization titers [19]. Using modeling, they demonstrated that neutralization level was highly predictive of immune protection. In another Israeli study, 39 HCWs from a large medical center with documented break-through infections and PCR data were studied [20]. Neutralizing antibody titers were significantly lower in case versus matched control patients during the peri-infection period. A new variant (B.1.1.7 (alpha)) was responsible for 85% of these breakthrough cases. Nearly three quarters of the patients had a high viral load at some point during their break-through infection and 19% had persistent symptoms (>6 weeks). A third report from University of California San Francisco (UCSF) described a decrease in vaccine effectiveness over time since HCWs received vaccinations [23]. While this study did not measure antibody response, it did measure infection rates in vaccinated vs. unvaccinated HCWs, which increased in both groups from March to July 2021. The authors hypothesized that this finding was related to both the circulating Delta variant and waning protection from vaccines over time.

Antibody responses post-SARS-CoV2 infection appear to be dependent on the illness severity [24,25]; there has been interest and speculation regarding vaccine effectiveness and its relationship to symptoms post-vaccination. Menni et al. analyzed self-reported symptoms post-vaccination using the COVID Symptom Study app in the UK and found that 25.4% of respondents reported one or more systemic symptoms [26]. Rechavi, in a study of 136 people in Israel, found a significantly higher antibody response a median of 35 days after the second dose of BNT162b in those who reported systemic reactogenicity [27]. Our study confirms this finding. We observed that HCWs with one or more symptoms reported after the second dose demonstrated a higher antibody response than those who reported no symptoms after the vaccine, and specifically, fatigue/tiredness, fever/chills, and body aches (Figure 1).

Our study has some limitations. We did not perform baseline anti-spike antibody tests prior to vaccination. However, we did perform baseline anti-nucleocapsid antibody testing as another aim of this study and reported previously that the infection rate was extremely low (1.3%) in a larger number of HCWs participating in early studies at our site [17]. Thus, any additive effect of previously infected individuals can be considered to have negligible impact to these results. Secondly, we did not record vaccine-related reactions at the time of vaccination but relied on the retrospective collection of this information from participants. We did not observe a difference in antibody levels by age or sex. Both European healthcare worker antibody studies found that antibody responses decrease with increasing age [11,12], and one of these two studies [12] found that women had higher antibody titers than men. These differences may reflect differences in our study population of younger, predominantly female participants, which were not accounted for in the regression analyses, or they could be due to other differences between European and American healthcare workers. Nevertheless, our study population reflects the age and gender distribution of frontline American healthcare workers, who are typically younger with a high representation of females [11]. Recent evidence suggests that antibody responses following vaccination may have greater durability in individuals with prior SARS-CoV2 infection [28]. The incidence of prior SARS-CoV2 infection in our HCW population was very low (<5%), thus this could not be evaluated in our study.

## 5. Conclusions

Understanding the factors that influence antibody response post-vaccination is important in determining recommendations for booster vaccinations. We documented that antibody concentration to the spike protein waned over seven months post-vaccination. Participants who reported greater reactogenicity post-vaccination had higher antibody concentrations. As single booster doses are now being recommended in the US, more studies are needed to determine what humoral response predicts protection from SARS-CoV-2 morbidity and mortality.

## Figures and Tables

**Figure 1 vaccines-10-00601-f001:**
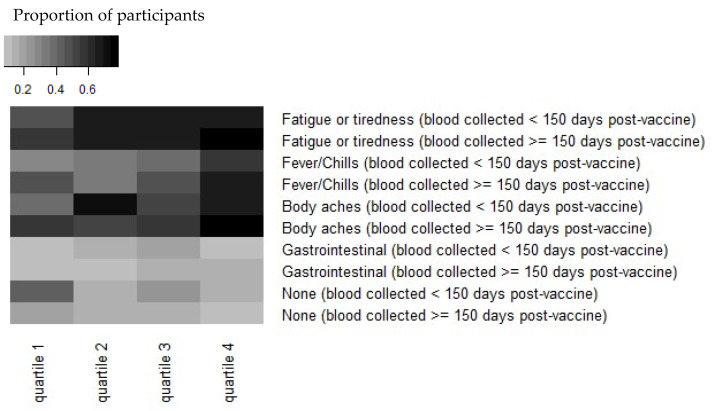
Heatmap demonstrating that reactogenicity post-vaccination is associated with increased Spike “S” antibody concentration. Antibody concentrations were stratified by quartiles in each timepoint and by symptoms at the time of vaccination. The quantitative IgG response to S quartiles for <150 days are: (1) 298–1440 AU/mL, (2) 1440–2606 AU/mL, (3) 2606–4198 AU/mL, and (4) 4198–16,283 AU/mL. The quartiles for ≥150 days are: (1) 166–1246 AU/mL, (2) 1246–1820 AU/mL, (3) 1820–2879 AU/mL, and (4) 2879–9278 AU/mL. Note that antibody concentrations were combined from symptoms reported before or after 150 days post-vaccination.

**Figure 2 vaccines-10-00601-f002:**
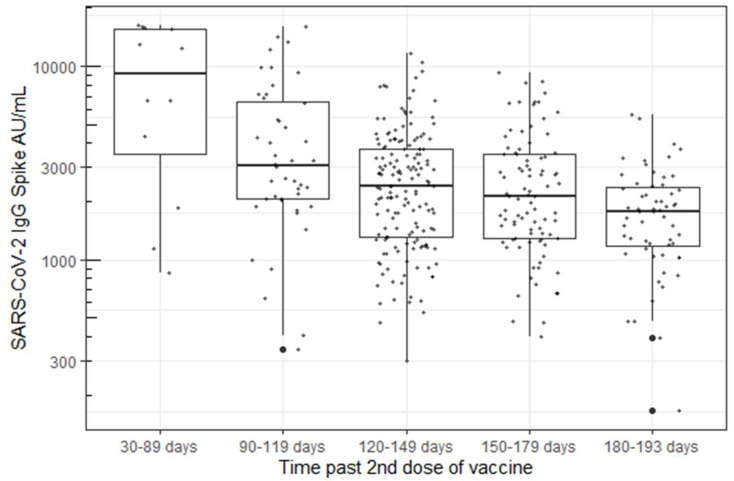
B. Decrease in Spike “S” IgG antibody concentration over time using the Abbott’s AdviseDx SARS-CoV-2 IgII assay.

**Table 1 vaccines-10-00601-t001:** Population Demographics.

Characteristic	Number (%)	SARS-CoV-2 IgG Spike (AU/mL) (Median [IQR])
*Vaccine*		
Pfizer BNT	363 (96)	2130.2 [1264.7, 3475.8]
Moderna mRNA-1273	13 (3.4)	7189.5 [4017.8, 9850.3]
I don’t recall	2 (0.5)	8450.45 [4637.7, 12263.2]
*Sex*		
Female	335 (86.6)	2173 [1243.45, 3712.1]
Male	52 (13.4)	2003.35 [1334.9, 3363.9]
*Age (years)*		
<35	164 (42.4%)	2418.4 [1355.65, 3721.45]
35–44	143 (37%)	1882.6 [1184.6, 3261.35]
45–55	52 (13.4%)	1852.65 [1225, 4370.3]
>55	28 (7.2%)	2045.35 [1349.3, 4018.25]
*Race*		
American Indian or Alaska Native	2 (0.5)	1240.7 [986.75, 1494.65]
Asian American	57 (14.7)	1940.7 [1429.4, 2740.8]
Native Hawaiian or other Pacific Islander	2 (0.5)	2287.25 [1759.7, 2814.8]
Black or African American	7 (1.8)	2838.8 [1406.9, 3859.65]
White	321 (82.9)	2247.5 [1242.6, 3789.6]
Other	5 (1.3)	1578.6 [4.5, 6662.6]
>1 race	10 (2.6)	2697.65 [1946.55, 3389.2]
*Ethnicity*		
Hispanic	25 (6.5)	1386.55 [882.8, 3484]
Not Hispanic	360 (93)	2173 [1272.8, 3586.1]
Prefer not to say	2 (0.5)	8942.1 [5375.15, 12,509]

**Table 2 vaccines-10-00601-t002:** Symptoms post-second-dose-vaccine, recorded at the fourth blood draw survey.

Symptom	Number (%)
None	76 (19.6)
Fever	130 (33.6)
Chills	137 (35.4)
Headache	169 (43.7)
Joint Pains	72 (18.6)
Muscle or body aches	214 (55.3)
Fatigue or tiredness	242 (62.5)
Nausea	44 (11.4)
Vomiting	5 (1.3)
Diarrhea	3 (0.8)
Abdominal pain	6 (1.6)
Rash (other than at the injection site)	3 (0.8)

## Data Availability

Not applicable.

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
