# Peer review of "Higher Antibody Concentrations in U.S. Health Care Workers Associated with Greater Reactogenicity Post-Vaccination"

_vaccines, 2022, doi:10.3390/vaccines10040601_

Round 1

Reviewer 1 Report

This study evaluated SARS-CoV-2 IgG spike antibody responses after the second  dose of vaccine in healthcare workers. The authors reported that antibody concentration to the spike protein decreased during the months following vaccination. Subjects who reported greater reactogenicity post-vaccination had higher antibodies concentrations.

The manuscript is clear and relevant for the field. The ethics statements are adequate. 

The limitations of the study are have been already highlighted by the authors:

1-absence of baseline anti-spike measurement

2- retrospective collection of vaccine-related reactions.

3- the study population is made predominantly by female participants.

I have a question regarding the collection of blood: I guess you used serum to perform the measurement, but I don’t understand why it has been refrigerated. Maybe it was not possible to store the blood quickly after collection: is it correct?

Did any subject experience a serious adverse event?

Author Response

Thank you for your thorough review of our paper. We appreciate the reviewer's comments and have addressed them below.

Comment 1: I have a question regarding the collection of blood: I guess you used serum to perform the measurement, but I don’t understand why it has been refrigerated. Maybe it was not possible to store the blood quickly after collection: is it correct?

That's correct, samples were centrifuged and refrigerated within 24 hrs of collection so that they could be stored until analysis.

Comment 2: Did any subject experience a serious adverse event?

No participants reported serious adverse events related to this study.

Additionally, at the request of the editor to increase word count, we've expanded a few areas and included 4 more references.

  • Lines 33-39, expanded background on vaccines.
  • Lines 61-64, 68-69, expanded description of similar studies in the literature
  • Lines 171-172, adding references on vaccine-associated immunity indicators
  • Lines 185-188, included an additional strength of this study: In addition, we had very high continued participation throughout the study (73% of the original cohort tested at the fourth visit) compared with other published studies, resulting in decreased opportunity for the introduction of bias in the population tested. 
  • Lines 190-193, additional description of Levin's study
  • Lines 194-203, description of newly added references on immune indicators: Two studies have demonstrated an association between SARS-CoV-2 antibody levels and immune protection. In a study by Khoury et al, the authors used published data from seven vaccine studies to determine the log mean of neutralization titers [19]. Using mod-eling, they demonstrated that neutralization level was highly predictive of immune pro-tection. In another Israeli study, 39 HCW from a large medical center with documented break-through infections and PCR data were studied [20]. Neutralizing antibody titers were significantly lower in case versus matched control patients during the peri-infection period. A new variant (B.1.1.7 (alpha)) was responsible for 85% of these breakthrough cases. Nearly three quarters of the patients had a high viral load at some point during their break-through infection and 19% had persistent symptoms (>6 weeks). 
  • Lines 236 - 239, added new reference that adds strength to our findings: Recent evidence suggests that antibody responses following vaccination may have greater durability in individuals with prior SARS-CoV2 infection [28]. The incidence of prior SARS-CoV2 infection in our HCW population was very low (<5%) so this could not be evaluated in our study.

Reviewer 2 Report

The manuscript by Dickerson et al. analyzed SARS-CoV-2 IgG spike antibody responses following the second dose of vaccine in healthcare workers. Overall, the experiments and results are straight-forward and the interpretation of the data agrees with the data as presented. The techniques are standard and appear to have been carried out appropriately.  It is not a novel concept but it represents one of relatively few reports that describe antibody concentrations beyond 3 months post-vaccination. Major suggestion : Symptoms recorded 0-5 months post second-dose vaccine. Table 2.  should be revised, the time frame of the symptoms must be defined more clearly. 0 to 5 month is an pretty big time frame and discussion of the reactogenicity should be revised accordingly.

Author Response

Thank you for your thorough review of our paper. We appreciate the reviewer's comments and have addressed them below.

Comment: Symptoms recorded 0-5 months post second-dose vaccine. Table 2.  should be revised, the time frame of the symptoms must be defined more clearly. 0 to 5 month is an pretty big time frame and discussion of the reactogenicity should be revised accordingly.

Reply: We agree that the labeling of Table 2 was not clear.  At the 4th blood draw survey, which occurred 50-193 days after the second vaccination, we asked participants about post-vaccination symptoms. We did not specify a specific time period for vaccine-related symptoms, and left it up to the participant to determine the time-frame for symptoms that they considered to be vaccine-related.  However, most reported symptoms were of short duration, as would be expected in the immediate post-vaccination period.  We also noted specifically in the discussion of limitations (line 198-199): Secondly, we did not record vaccine-related reactions at the time of vaccination but relied on the retrospective collection of this information from participants.  

We have updated the Table to read: Symptoms post second-dose vaccine, recorded at the 4th blood draw survey. 

We also updated the methods (lines 103-104) to read: At the time of the 12-month post-enrollment blood draw, participants were also asked to complete a survey, based on CDC’s v-safe survey [17],  documenting the number and severity of symptoms experienced after the second dose of the vaccine (Table 2).

And we added lines 109-111: Because participants were vaccinated at different times, this post-vaccination specimens could have been obtained anywhere from one to six months after vaccination. 

Additionally, we clarified the results to read (line 148-149): Reactogenicity experienced immediately after the second dose of vaccine was reported by 311 (80.4%) participants at the 4th blood draw survey, a median of 136 days post vaccine dose 2 (Range, 50-193 days). 

Finally, in the discussion, we clarified in lines 218-220: We observed that HCWs with one or more symptoms reported after the second-dose demonstrated a higher antibody response than those who reported no symptoms after the vaccine, and specifically, fatigue/tiredness, fever/chills, and body aches (Figure 1). 

Round 2

Reviewer 2 Report

The Authors addressed all of the reviewers comments in adequate manner.